# Associations between socioeconomic position and changes in children's screen-viewing between ages 6 and 9: a longitudinal study

Ruth E Salway [iD],[1] Lydia Emm-Collison,[1] Simon Sebire,[1] Janice L Thompson,[2] Russ Jago [iD] [1]

¹Centre for Exercise, Nutrition and Health Sciences, School for Policy Studies, University of Bristol, Bristol, UK
²School of Sport and Exercise Sciences, The University of Birmingham, Birmingham, UK

**Correspondence to**
Ruth E Salway;
ruth.salway@bristol.ac.uk

## ABSTRACT

**Objectives** To explore socioeconomic differences in screen-viewing at ages 6 and 9, and how these are related to different media uses.

**Design** Longitudinal cohort study.

**Setting** Children recruited from 57 state-funded primary schools in Southwest England, UK.

**Participants** 1299 children at ages 5–6, 1223 children at ages 8–9, including 685 children at both time points.

**Outcome measures** Children's total screen-viewing time (parent-reported) and time spent using multiple screen devices simultaneously (multiscreen viewing), for weekdays and weekends.

**Methods** Negative binomial regression was used to model associations between socioeconomic variables (highest household education and area deprivation) and total screen-viewing at age 6 and the change from age 6 to 9. We additionally adjusted for child characteristics, parental influences and media devices in the home. Multiscreen viewing was analysed separately.

**Results** Household education was associated with children's screen-viewing at age 6 with lower screen-viewing in higher socioeconomic groups (21%–27% less in households with a Degree or Higher Degree, compared with General Certificate of Secondary Education: GCSE). These differences were explained by the presence of games consoles, parental limits on screen-viewing and average parent screen-viewing. Between ages 6 and 9, there were larger increases in screen-viewing for children from A level and Degree households (13% and 6%, respectively, in the week) and a decrease in Higher Degree households (16%), compared with GCSE households. Differences by household education remained when adjusting for media devices and parental factors.

**Conclusions** Children's screen-viewing patterns differ by parental education with higher levels of viewing among children living in households with lower educational qualifications. These differences are already present at age 6, and continue at age 9. Strategies to manage child sedentary time, and particularly screen-viewing, may need to take account of the socioeconomic differences and target strategies to specific groups.

## Strengths and limitations of this study

► Longitudinal study screen viewing on both weekdays and weekends at two time points.
► Time frame covers an important period of change in child-viewing behaviours.
► Data collection period coincided with a period of rapid change in screen-viewing technology which has facilitated the presentation of contemporary data.
► We did not explicitly ask parents about tablet or smartphone use and so have not been able to capture this directly.
► A change in the way education was recorded may result in misclassification in a small number of situations where household composition has changed or where people have obtained additional qualifications.

## INTRODUCTION

Sedentary behaviour has been associated with increased risk of obesity and elevated levels of cardiometabolic risk factors among children and adolescents.[1 2] Screen-viewing is a very common form of sedentary behaviour among children and young people,[3] and higher screen-viewing has been associated with higher body mass index (BMI), poorer psychological well-being and poorer academic achievement among children.[4–6] The American Academy of Paediatrics recommends restricting children's screen-viewing, although it does not specify a limit.[7] An international study estimated that 11-year old boys engaged in an average of 5.3 h/day of screen-viewing on weekdays in 2010 (4.4 h/day for girls) with higher estimates at weekends.[8] The paper therefore shows that by the end of primary school, children are engaging in high levels of screen viewing. Sedentary behaviour tracks at moderate levels from childhood into adolescence,[9] with TV viewing tracking

from childhood to adulthood.[10] Moreover, screen-viewing has increased in children by around 1.3 h/day on weekdays (2.0 h/day on weekends) between 2002 and 2010.[8] Thus, screen-viewing is an increasingly common behaviour among children and with levels increasing as children age. We therefore need to understand patterns of behaviour develop during childhood, and particularly at the start of primary school when patterns of behaviour are established. As such, information on levels of screen-viewing and how they change during early school years is required.

Much of the current literature on screen-viewing focuses on TV viewing, but screen-viewing has evolved with technology to include TV viewing, computer games, tablets, mobile phones and multiscreen viewing (in which children use two or more devices at the same time),[11] and become more diverse, for example encompassing on-demand TV and online gaming.[3] Thus, to try and change behaviour, there is a need to understand what devices children are using and the types of viewing in which children engage on different days of the week.

Socioeconomic position has been associated with poorer health outcomes among children and young people. It is important to note, however, that different indicators of socioeconomic position, such as education, income and deprivation, measure different, often related aspects, although they are strongly correlated.[12 13] For example, in England at the end of primary school (age 11) in 2018, the prevalence of child obesity in the most deprived areas is more than double that of those in the least-deprived neighbourhoods (27% vs 12%),[14] and TV viewing has been identified as a mediator for this association.[15] A number of studies have found an inverse relationship between socioeconomic status (SES) and sedentary time,[16] sedentary behaviours[17 18] and screen-viewing specifically[19 20] with children's daily screen time varying between 1.7 and 2.4 hours/day for high and low SES families, respectively.[19] Sedentary behaviour increases as children age,[18] but as much of the evidence is cross-sectional, it is not clear whether different SES groups change at the same rate. Longitudinal evidence between ages 2 and 9 suggests that while TV viewing increases similarly for different educational subgroups, there may be different trajectories for different household income groups.[21] It is also not clear whether these SES differences are due to differences in other factors or not. Child obesity, ownership of different media devices, parental screen-viewing and parental limits on screen-viewing have all been found to affect children's screen-viewing and differ between SES groups.[1 19 22–29] There is thus a need to better understand SES differences, and especially how these change over time.

The aim of this paper was to explore socioeconomic differences in screen-viewing at age 6, and the change in screen-viewing between ages 6 and 9, and whether any differences can be explained by other factors such as different media devices and parental influences. Developing knowledge in this area will support the creation of targeted behaviour change programmes to reduce screen-viewing.

## METHODS

B-PROACT1V is a longitudinal study that aimed to examine the physical activity and sedentary behaviours of primary school children aged 5–11 years, and their parents (described in detail elsewhere[26 30 31]). In phase 1, all children in Year 1 of primary school (aged 5–6 years) from 57 schools in and around Bristol were invited to participate, with data collected between January 2012 and July 2013. In phase 2, when the children were in Year 4 (aged 8–9 years), all schools from phase 1 were invited to participate, with 47 schools agreeing. All children were eligible regardless of whether they had participated in phase 1, and data collection took place between March 2015 and July 2016. Data were collected for 1299 children in Year 1 and 1223 children in Year 4, with 685 children included in both phases. Self-identified 'first' parents completed a questionnaire about personal and family characteristics while 'second' parents completed a shorter questionnaire. This paper uses data from the first parent questionnaires: 1085 (84%) from Year 1, 997 (82%) from Year 4 and 509 (74%) at both time points.

### Screen-viewing data

In both years, the first parent was asked about the number of hours their child typically spent engaging in specific screen-viewing behaviours on weekdays and at weekends: TV, computer and games consoles. Additionally, both parents were asked about their own screen-viewing. These responses were recorded as either 'None' or in hourly categories from '0–1 hours' up to '4 hours or more', and recoded based on midpoints to give the average number of minutes spent in each type of screen-viewing on weekdays and weekends. These were summed to form the average total number of minutes spent on any type of screen-viewing. Where two parents completed questionnaires, parental screen-viewing was taken as the average (approximately 27% of respondents in each year). At the second phase, parents were also asked about the time they and their child spent multiscreen-viewing (ie, using two or more devices at once), with responses as above recoded to midpoints.[11]

### Socioconomic data

At age 6, the first parent was asked their highest educational qualification, while at age 9, they were asked the highest education qualification of anyone in the household. We combined these to form the highest household educational qualification recorded at either time with categories 'Up to GCSE (General Certificate of Secondary Education) or equivalent' (qualification at age 16), 'A level/National Vocational Qualification or equivalent' (qualification at age 18), 'University Degree or equivalent' and 'Higher Degree (MSc/PhD) or equivalent'. In addition, Index of Multiple Deprivation (IMD) scores,

based on the English Indices of Deprivation (http://data. gov.uk/dataset/index-of-multiple-deprivation), were assigned to each child based on their reported home postcode, with higher IMD scores indicating a greater level of deprivation. Household education captures individual long-term socioeconomic position, such as knowledge, and also provides an indirect indication of income, while IMD captures the socioeconomic conditions of the area in which they live.

## Other measurements

Child gender was reported by the first parent. Child height and weight were recorded to the nearest 0.1 cm and 0.1 kg, respectively, by trained fieldworkers at each time point. BMI was calculated and converted to an age-specific and sex-specific SD score based on UK reference curves.[32 33] The first parent was asked whether they limited the time their child spent engaging in three different types of screen-viewing (TV, computer and video games) with responses from 1 'Strongly disagree' to 4 'Strongly agree'. The average of these was used to capture parental limits on screen-viewing. We also asked about the number of media devices in the home: TVs, computers (desktop or laptop), tablet computers and games consoles (including handheld consoles).

## Patient and public involvement

The research questions for this study emerged from research that was conducted with 1078 Year 1 children. A subset of 53 parents of the children who took part in the Year 1 study participated in interviews to help guide future research. Participants provided verbal assent (with parental consent) to join the study, and summaries of project findings are sent to all participants via study schools.

## Statistical analysis

Descriptive summaries of children's screen-viewing on weekdays and weekends and other participant characteristics were produced by household education group for ages 6 and 9. To aid interpretation, the screen-viewing variables are reported in summaries as continuous variables. However, the underlying variables are discrete and so in the main analyses, we used negative binomial regression models (similar to Poisson models but suitable for over-dispersed count data) to model weekday and weekend screen-viewing separately. We considered four main models. Model 1 explored cross-sectional associations between socioeconomic position (household education and IMD) and total minutes of screen-viewing at age 6. Model 2 additionally adjusted for possible mediators and confounders: child gender, child BMI, number of devices in the home (more than one TV, and presence of any computers, tablets or games consoles), parental screen-viewing and parental behaviour on limiting screen-viewing. Models 3 (unadjusted) and 4 (adjusted for covariates) explored the longitudinal change between ages 6 and 9; that is, associations between socioeconomic

position and total minutes of screen-viewing at age 9, adjusting for baseline screen-viewing at age 6. Finally, we also explored the association between socioeconomic position and multiscreen viewing as a separate outcome. Robust standard errors were used to account for clustering of children within schools, and all analyses are based on complete cases. Model results are presented as screen time ratios (rate ratios), defined as the exponent of the model coefficients. An increase of one unit in a given predictor variable is associated with a multiplicative increase in screen-viewing, holding the other predictor variables in the model constant. Model assumptions were checked via model diagnostics, and all analysis was done in Stata V.15.

## RESULTS

Children's screen-viewing differed between households with differing levels of education at both ages 6 and 9, with higher screen-viewing in lower qualified households, decreasing as education increased, with 35–51 min difference between the highest and lowest household education qualifications (figure 1). Time spent screen-viewing was higher at age 9 than age 6, and higher at weekends than during the week among all education groups, with time spent watching TV and playing games consoles nearly doubling at weekends (online supplementary table S1 and S2). Time spent engaged in multiscreen viewing was also higher for households with lower levels of qualification, with children in households where GCSEs were the highest qualification spending over twice as much time in multiscreen viewing as those where the highest qualification was a postgraduate degree, on both weekdays and weekends. Missing data ranged between 0% and 3% (online supplementary table S3).

There were differences in the number and types of media devices in the home by household education (figure 2). Households where GCSE and A Levels were the highest qualification had more TVs and games consoles in the house, while Higher Degree households had more computers. There were small increases in most devices between ages 6 and 9. The average number of tablet devices was the same across all levels of household education but increased fourfold to an average of 2.2 per household between the two assessment periods. Parental screen-viewing differed with household education by similar amounts to their children, again with higher screen-viewing among those with lower level qualifications. Parental screen-viewing was also higher at the second time point, especially among the Higher Degree educated households, and differences with education were smaller.

Table 1 shows the associations between socioeconomic position and total screen-viewing at age 6, in terms of ratios for screen time. For example, a factor of 0.79 for Degree-level household education (table 1, weekday) means that screen-viewing time decreased by 21% for children in households where a University degree or

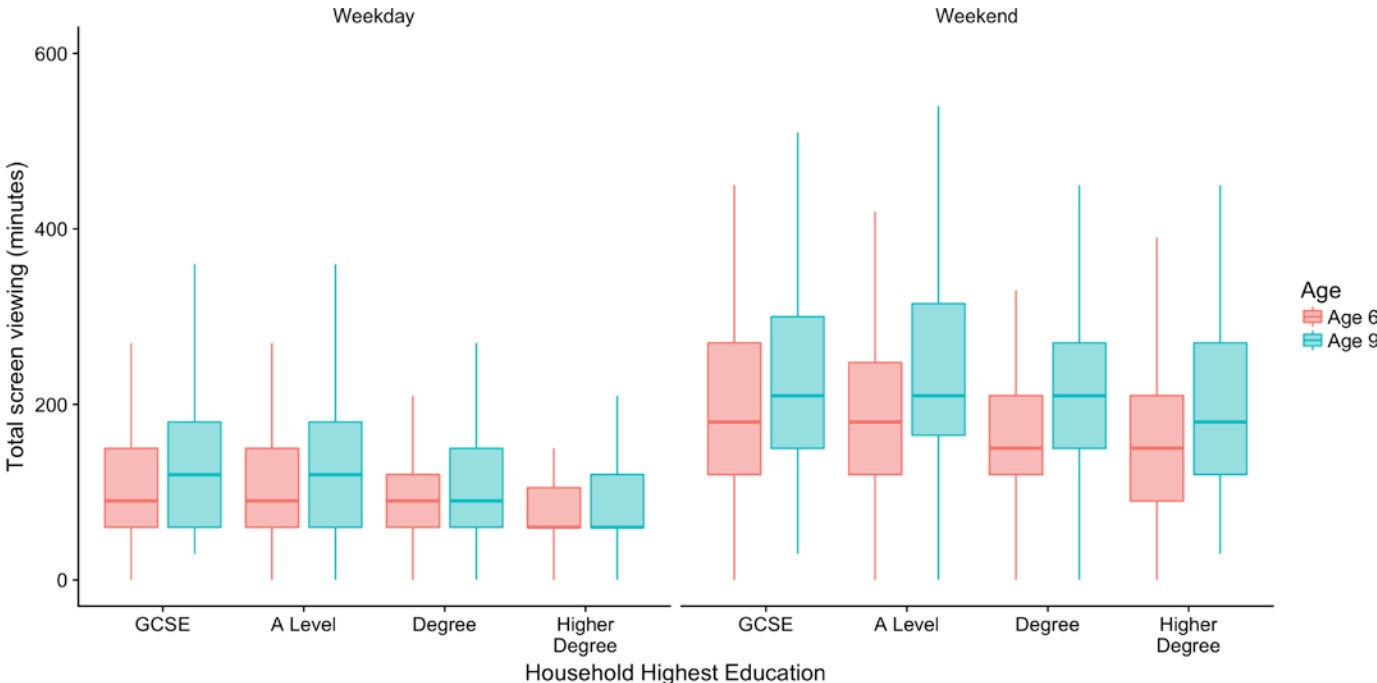

**Figure 1** Boxplots of children's total minutes screen-viewing by household education at ages 6 and 9, for weekdays (left) and weekends (right). GCSE, General Certificate of Secondary Education.

equivalent was the highest qualification, compared with children in households where the highest qualifications were GCSEs. In the unadjusted model (Model 1), there were differences in screen-viewing by household education for both weekdays and weekends, with lower screen-viewing for children in households with higher education. The GCSE and A level education groups were similar, but children in households with a Degree or Higher degree engaged in 21%–27% less screen-viewing.

There was no association with IMD. The adjusted model (Model 2) adjusts for possible mediators: number of devices, child BMI, parental screen-viewing and parental limits on screen-viewing all measured at age 6. These factors accounted for household education differences on both weekdays and weekends. Children's screen-viewing was higher in households with games consoles, and lower when parents were more likely to limit screen-viewing (online supplementary table S4). There were

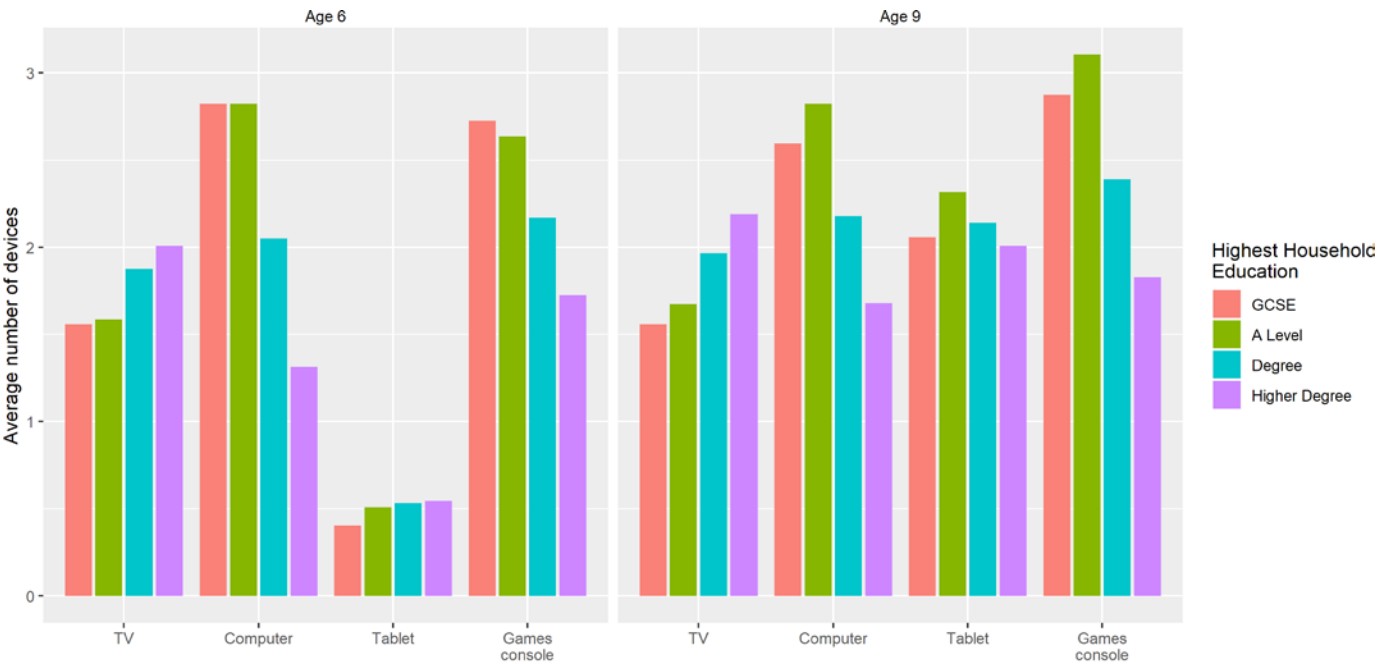

**Figure 2** Average number of devices in the home by household education at ages 6 (left) and 9 (right). GCSE, General Certificate of Secondary Education.

**Table 1** Cross-sectional associations between socioeconomic position and screen-viewing at age 6

| | Model 1: unadjusted | | | Model 2: adjusted* | | |
|---|---|---|---|---|---|---|
| | Ratio | 95% CI | P value | Ratio | 95% CI | P value |
| Weekday | | n=1043 | | | n=953 | |
| Household education | | | | | | |
| Up to GCSE | 1 | (Reference) | | 1 | (Reference) | |
| A level | 0.94 | (0.85 to 1.03) | | 0.98 | (0.89 to 1.08) | |
| Degree | 0.79 | (0.70 to 0.90) | | 0.88 | (0.78 to 1.00) | |
| Higher degree | 0.73 | (0.60 to 0.88) | <0.0005† | 0.88 | (0.74 to 1.05) | 0.054† |
| Deprivation (IMD)‡ | 1.04 | (1.00 to 1.08) | 0.067 | 1.02 | (0.98 to 1.07) | 0.296 |
| Weekend | | n=1038 | | | n=946 | |
| Household education | | | | | | |
| Up to GCSE | 1 | (Reference) | | 1 | (Reference) | |
| A level | 1.00 | (0.91 to 1.10) | | 1.03 | (0.95 to 1.12) | |
| Degree | 0.88 | (0.79 to 0.98) | | 1.02 | (0.93 to 1.13) | |
| Higher degree | 0.77 | (0.67 to 0.90) | 0.001† | 0.99 | (0.87 to 1.11) | 0.725† |
| Deprivation (IMD)‡ | 1.02 | (0.98 to 1.06) | 0.387 | 1.01 | (0.96 to 1.06) | 0.720 |

*Adjusted for child gender, child BMI, presence of TVs, computers, tablets and games consoles in the household, parental screen-viewing and parental limiting of screen-viewing.
†P-value: test for differences between education categories.
‡Increase per 1 standard deviation (14.0) in IMD: higher values indicate more deprived areas.
GCSE, General Certificate of Secondary Education; IMD, Index of Multiple Deprivation.

**Table 2** Longitudinal associations between socioeconomic position and screen-viewing at age 9, adjusting for baseline screen-viewing at age 6

| | Model 3: unadjusted | | | Model 4: adjusted* | | |
|---|---|---|---|---|---|---|
| | Ratio | 95% CI | P value | Ratio | 95% CI | P value |
| Weekday | | n=487 | | | n=476 | |
| Household education | | | | | | |
| Up to GCSE | 1 | (Reference) | | 1 | (Reference) | |
| A level | 1.13 | (1.01 to 1.26) | | 1.16 | (1.03 to 1.32) | |
| Degree | 1.06 | (0.92 to 1.22) | | 1.10 | (0.96 to 1.26) | |
| Higher degree | 0.84 | (0.70 to 0.99) | 0.001† | 0.90 | (0.77 to 1.05) | 0.008† |
| Deprivation (IMD)‡ | 1.06 | (1.00 to 1.13) | 0.053 | 1.05 | (1.00 to 1.10) | 0.038 |
| Weekend | | n=483 | | | n=469 | |
| Household education | | | | | | |
| Up to GCSE | 1 | (Reference) | | 1 | (Reference) | |
| A level | 1.14 | (1.00 to 1.30) | | 1.14 | (1.01 to 1.28) | |
| Degree | 1.09 | (0.96 to 1.23) | | 1.11 | (0.99 to 1.24) | |
| Higher degree | 0.96 | (0.83 to 1.10) | 0.004† | 0.98 | (0.87 to 1.11) | 0.006† |
| Deprivation (IMD)‡ | 1.02 | (0.98 to 1.07) | 0.346 | 1.00 | (0.95 to 1.05) | 0.938 |

*Adjusted for child gender, presence of TVs, computers, tablets and games consoles in the household, parental screen-viewing and parental limiting of screen-viewing.
†P-value: test for differences between education categories.
‡Increase per 1 standard deviation (14.0) in IMD: higher values indicate more deprived areas.
GCSE, General Certificate of Secondary Education; IMD, Index of Multiple Deprivation.

weak associations with parental screen-viewing and child BMI.

Models 3 (unadjusted) and 4 (adjusted) examined the change in screen-viewing between ages 6 and 9 (table 2). In Model 3, adjusting for baseline screen-viewing, there were differences in screen-viewing by household education, with larger increases in screen-viewing for children from A level and Degree households (13% and 6%, respectively) and 16% decrease in Higher Degree households, compared with GCSE households in the week. Similar patterns were seen at weekends, with the A level and Degree groups increasing more than the GCSE and Higher Degree groups. There was no association with IMD. In Model 4, when adjusting for media devices in the home, child BMI and parental influences at age 9, household education differences still remain, with larger increases for the A level and Degree groups. An increase in screen-viewing was strongly associated with the presence of games consoles and multiple TVs (weekdays only), with weaker associations with parental screen-viewing (online supplementary table S5). Presence of tablets was associated with a decrease in screen-viewing, as was parental limits on screen-viewing in the week, but not at weekends.

Multiscreen viewing was associated with both IMD but not with household education, with children's multiscreen viewing higher in families living in more deprived areas (online supplementary table S6 and figure S1). When adjusting for other factors, the association with deprivation remained, with multiscreen viewing 26%–27% higher for every SD increase in IMD score. Higher amounts of multiscreen viewing were associated with higher levels of parental multiscreen viewing and the presence of tablet devices and multiple TVs in the week, and lower levels of multiscreen viewing were associated with increased parental limits on screen-viewing.

## DISCUSSION

The data presented in this paper have shown an association between children's screen-viewing and highest household education level, with higher screen-viewing in lower socioeconomic groups. These differences were already present at age 6 and continued to be evident at age 9. Socioeconomic differences at age 6 were accounted for by other factors, especially parental factors such as their own screen-viewing and their behaviour in limiting their children's screen-viewing. However, while the time children spent engaged in screen-viewing increased in all households between ages 6 and 9, regardless of education, they increased by different amounts depending on household education. Adjusting for baseline screen-viewing at age 6, screen-viewing increased most over the time period for children in households where the highest qualifications were A Levels or a university degree compared with GCSE, with the smallest increase among Higher Degree households. In A-level educated households, this reflected a

'catching-up' with the level of screen-viewing seen in GCSE-educated households. These findings imply socioeconomic differences in both amount of screen-viewing, and the way in which screen-viewing patterns develop. These differences were related to household education rather than area-level deprivation, suggesting that it is long-term socioeconomic aspects such as parental knowledge, ability to engage and communicate with services, and possibly income that are important, rather than the socioeconomic conditions of the area, such as availability of local resources. In contrast, multiscreen viewing is associated with area deprivation rather than education and so could represent a proxy for short-term current income or possibly a neighbourhood effect of what is considered to be 'typical' screen-viewing behaviour. As such, strategies to manage child sedentary time, and particularly different types of screen-viewing, may need to take account of the socioeconomic differences and target strategies to specific groups.

Ownership of media devices such as TVs, phones and games consoles, has previously been found to be associated with higher levels of screen-viewing, and also inversely with SES.[24 28] We found that the number of media devices in the home differed by household education, with more TVs and games consoles in households with GCSE and A level qualifications, and more computers in households with a university or higher degree. While the number of devices changed overall between the two assessment points, most notably with a large increase in tablet devices and a slight decrease in the number of computers, socioeconomic differences were small and did not account for the observed differences in total screen-viewing between household education. We found that higher screen-viewing at age 6 was associated with games consoles at the weekends only, but not with multiple TVs or computers, and larger increases in screen-viewing between ages 6 and 9 were associated with games consoles and multiple TVs in the week. However, we also saw tablet ownership associated with smaller increases in screen-viewing between ages 6 and 9. This suggests that the relationship between different devices and screen-viewing is complex, and may be changing over time. Thus, we extend previous findings to show that while access to devices may contribute to baseline screen-viewing, it does not entirely explain socioeconomic differences in the increase in screen-viewing between ages 6 and 9.

There were weak associations between parent and child screen-viewing, with every 30 min of parental screen-viewing associated with a 4% increase in child screen-viewing. Parental limits on screen-time were strongly associated with lower screen-viewing at age 6 and less time spent in multiscreen-viewing at age 9, but associations were stronger for weekdays than weekends and limiting screen time was associated with changes in screen-viewing between ages 6 and 9 on weekdays only. Interestingly, analysis of interviews conducted with a subsample of the parents in the study showed that many parents are uneasy about managing non-TV screen-time and feel that they

struggle to keep up with rapid technological change.[34] Collectively, these findings may suggest that there is a need to help parents to identify effective ways to manage constantly adapting forms of viewing.

Tablet ownership did not vary with household education but increased greatly overall between ages 6 and 9. A limitation of this project is that we did not assess tablet or smartphone use as a specific behaviour in the questionnaire. This may underestimate total screen-viewing, especially for those who are heavy users of these devices, and may explain the observed negative association between screen-viewing and tablet ownership. We also note that tablet ownership is strongly associated with higher levels of multiscreen-viewing on weekdays. The widespread ownership of tablets is particularly important for understanding the complexity of screen-viewing, as watching TV, playing games and using the internet can all be done via a tablet and, as such, identifying the behaviour engaged in while using a tablet is challenging. Thus, it may be that in the future there is a specific need to assess tablet time, and to differentiate the various ways in which tablets are used. For example, applications that allow parents and children to agree targets to limit the time spent on games and other non-educational activities could also be used as a feedback and monitoring device.

The data presented in this paper add to the previous evidence base which has shown that socioeconomic position is associated with sedentary time and screen-viewing. Previous work within this area has focused on cross-sectional associations between TV time and various indicators of socio-economic position.[15–20] Collectively, this body of evidence identifies an inverse association with children residing in more socioeconomically disadvantaged households engaging in more TV viewing. The findings from this paper greatly extend this work by showing that these patterns are existent cross-sectionally and prospectively. The prospective associations, are however, particularly informative as they show that these differences are evident at age 6 and continue as child ages. These findings suggest that we need a range of options to help families to manage their screen-time and that these made need to be tailored to specific socioeconomic groups. For example, for families from lower socioeconomic groups, there is likely to be a need for programme that focus on stopping the development of screen-viewing behaviours in early childhood. This could be parental education programme that are combined with behaviour change techniques, while for older children from all socioeconomic groups, there may need to be more of a focus on family-focussed management information where children and parents work together to set family viewing goals.[35] It is unlikely that a single strategy will work on its own,[36] and as such there is need for multiple strategies across childhood that adapt to the age and viewing habits of the child and the family.

The major strengths of this study are the information on a variety of different types of screen-viewing at two time points which span an important period of change in child-viewing behaviours. The data collection period (2012–2016) coincided with a period of rapid change in screen-viewing technology which has facilitated the presentation of contemporary data. There are, however, several limitations that need to be considered. First, we did not explicitly ask parents about tablet or smartphone use. As data collection covers a time when screen use is changing rapidly (eg, global tablet sales increased from 116 million in 2012[37] to 207 million in 2015[38]), this is an important aspect we have not been able to capture directly. Second, the differences in the education variable between Year 1 (first parent's education) and Year 4 (household education) may result in inaccuracies in situations where household composition has changed or where people have obtained additional qualifications between the two time points, although we believe that the number of these cases is likely to be small. Children's screen-viewing was parent-reported, and amounts of multiscreen-viewing, in particular, may not be well-captured, although general patterns might be more robust. This is a longitudinal study, and so some children were lost to follow-up, although the majority were through non-participating schools. In addition, as we have included socioeconomic position variables directly, our models will not biassed due to differential follow-up rates by SES. Finally, as parental screen-viewing increased between 2012 and 2015, this may indicate a general change in viewing patterns, and the increases reported between the two age periods could reflect secular changes as opposed to age-related differences. As such, it would be important to identify if comparable patterns are evident in other data sets.

## CONCLUSION

Children's screen-viewing patterns differ by parental education with higher levels of viewing among children living in households with lower levels of education. These differences are already present at age 6, and continue at age 9, even when accounting for differences in baseline screen-viewing and device ownership. Socioeconomic differences narrow with age as children in households with higher qualifications gain greater access to screen-viewing devices. Strategies to manage child sedentary time, and particularly screen-viewing, may need to take account of the socioeconomic differences and target strategies to specific groups. For example, specific early intervention strategies to reduce screen-viewing in children from lower socioeconomic groups are likely to help stop high levels of screen-viewing from developing and reduce inequalities.

**Acknowledgements** We would like to thank all of the families and schools that have taken part in the B-PROACT1V project. We would also like to thank all current and previous members of the research team who are not authors on this paper.

**Contributors** RJ, SS and JLT were involved in the design of this study and in seeking funding for it. The paper was conceived by RJ and RES, and RES performed all analyses. RES and RJ wrote the first draft of the paper and RJ coordinated contributions from other authors. All authors (RJ, RES, SS, JLT and LE-C) made critical comments on drafts of the paper.

**Funding** This work was supported by the British Heart Foundation (ref PG/11/51/28986 and SP 14/4/31123). The funder had no involvement in data analysis, data interpretation or writing of the paper.

**Competing interests** None declared.

**Patient consent for publication** Not required.

**Ethics approval** The study received ethical approval from the School of Policy Studies Ethics Committee at the University of Bristol, UK, and written parental consent was received for all participants (Reference B-Proact1V study, approval 11 November 2011.)

**Provenance and peer review** Not commissioned; externally peer reviewed.

**Data availability statement** Data from the B-Proact1v study are in the process of being prepared for archiving. We will consider reasonable requests for access to the data once this is complete from September 2020.

**ORCID iDs**
Ruth E Salway http://orcid.org/0000-0002-3242-3951
Russ Jago http://orcid.org/0000-0002-3394-0176

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
