## [Reviewer comments · BMJ Open]

ARTICLE DETAILS

TITLE (PROVISIONAL)	Associations between socio-economic position and changes in children's screen-viewing between ages 6 and 9: a longitudinal study
AUTHORS	Salway, Ruth E; Emm-Collison, Lydia; Sebire, Simon; Thompson, Janice L; Jago, Russ

VERSION 1 – REVIEW

REVIEWER	Lilian Krist Charité-Universitätsmedizin Berlin
REVIEW RETURNED	06-Dec-2018

GENERAL COMMENTS	The authors of "Socio-economic patterns of changes in children's screen-viewing between ages 6 and 9: a longitudinal study" address the question of how socioeconomic differences influence changes in screen-viewing time among children between 6 and 9 years. I have the following remarks for the authors to consider: Major: The content of the paper is not easy to follow. The presentation of the results is only partial, e.g. increases of screen viewing time is described only for three out of four socioeconomic subgroups. The figures (especially Figure 2) are not very comprehensible, too. Title: What do the authors mean with "socioeconomic patterns"? Why don't they use "status". They don't describe any patterns. The title does not correspond well with the aim described in the introduction. Abstract: Please don't use abbreviations in the abstract or at least explain them when used for the first time. The authors present results in the conclusion section that they did not mention in the results section. Please change that. Introduction: The introduction section is very short with weak information about the existing body of knowledge. (Percentage of screen time of overall sedentary time in comparable population groups, recommendations regarding sitting behaviours...) The authors mix sedentary behaviour with screen viewing time and cite results of studies that don't fit very well into the context. (p6, 110-14). Methods:
--

	Please explain the educational levels. If abbreviations are used for the first time, they have to be explained. Please explain "GCSE", "NVQ" and "HND". Were height and weight measured at school or by the parents? Results: The first main result is that screen viewing is higher among children of parents with lower education. This is described in Table 1 and 2 where it is comprehensible. It is also described in Figure 1, if I understand correctly. Please explain though, what you presented with Figure 2. The title is again "total screen viewing", however it is not explained what the authors mean with "modelled screen viewing". Are these changes of screen viewing? Where are the numbers presented in this figure in the text? For example: Figure 1 (right) shows about 240minutes of screen time (weekend) at age 9 in the GCSE group. Figure 2 (left) shows 190 minutes of screen time (weekend) at age 9 in the GCSE group. Please explain that. Titles of figure 1 and 2 are misleading, please clarify. The authors present only the changes of the A Level group and the university degree group compared to the GCSE group without mentioning the higher degree group. Why is that? I think it is worth being mentioned that between the group with the lowest and the highest education level there were no differences in screen time changes. Please go more into detail when presenting these results. Discussion: When describing the main results, please mention all education groups! Page 17, line 50: The presented results belong to the results section. Page 18, line 8: the number of devices changed in two directions. Less computers but more tablets. Please differentiate. Page 18, line 15: The authors present a rate reduction of screen time for having any tablets at home. Thus, the conclusion that the presented results are consistent with previous studies (number of devices is linked with screen time) is false. The result of the present study is contradictory to that. The authors discuss different activities that tablets can be used for, however, it was not the aim of the study to differentiate between different screen activities but between different devices. Overall, the structure of the manuscript is not straight forward and should be revised.
--	---

REVIEWER	Zdenek Hamrik Faculty of Physical Culture, Palacky University Olomouc, Czech Republic
REVIEW RETURNED	22-Feb-2019

GENERAL COMMENTS	First of all, I would like to thank you for the opportunity to review the manuscript entitled "Socioeconomic patterns of changes in
---

	children’s screen-viewing between ages 6 and 9: a longitudinal study". The manuscript asses different types of screen viewing in the context of age and socio-economic differences based on data from B-PROACT1V longitudinal study. Generally, I like the idea of the paper, however, I have several comments dealing with the study that should be addressed. I hope it will be helpful for improving the overall quality of the paper. General comments I have 3 main issues that I think should be addressed:  - I would recommend simplifying general storyline dealing with the main aim, results and discussion section. Sometimes it was difficult to follow and recognize main messages - I would emphasize and discuss more in detail the issue of not including tablets and smartphones - It was not clear to me how multiscreen viewing was assessed or calculated Introduction General comment I would recommend adding to the text why the period between 6 and 9 is important for developing/fixing screen-based behavior Page6/line9-18 I do not clearly understand the storyline of the introduction here with the ref. no 16. I think this part should be better structured. Methods Page6/line47 Could you please add the No. and the date of ethical approval? Page6-7 I would recommend adding more detailed information about the response rate. For example, what was the response rate on school level in Phase 1? What was the response rate on an individual level, how many children refused to participate? What were the main reasons for not participating in the study? Page7/8-11 Could you please add the information about sex, I think this should be mentioned in the text not only in tables? page7/line26 Could you please explain how multiscreen viewing was assessed or calculated? The study of Jago 2011 that you refer to (ref. 26) stated that “there are currently no means of assessing multi-screen viewing and therefore techniques that assess multi-screen viewing are urgently needed”. It seems to clarify this in the study. page7/line47-52 In terms of the international character of the journal I would recommend explaining the abbreviation dealing with the household education qualification Page9/line 26-34 Imputation – I would consider to state only for example: “no imputed data were included in the study” and delete the rest Results Page9/line 50-55 I would not recommend starting the first sentence of the results section like this – please consider to change it. Moreover,
--	---

	considering the main aim, I don't think that the storyline of the results should start with the differences between weekdays and weekend. Page 9/first paragraph Please consider adding the information about the age like... "At the age of 6... children..." Page10/line8 I would recommend skipping the sentence: The percentage of sedentary... Discussion I would recommend to follow the same storyline from the results section and change the discussion accordingly. Limitations Authors stated that the main limitation of the study is the absence of assessing "tablet time". It seems to me that also measuring time spent on smartphones might be important for the results and should be mentioned in the limitation of the study. I would also recommend that the potential influence of the results dealing with this should be better discussed. Abstract Please change according to previous comments.
--	--

REVIEWER	Augusto Di Castelnuovo Mediterranea Cardiocentro, Napoli (Italy)
REVIEW RETURNED	30-Apr-2019

GENERAL COMMENTS	The authors stated they "...not presented any imputed data in the paper", because they encountered problems with convergence in applying multiple imputation methods. However, it is not clear how they consequently dealt with missing data. Since they used multivariable negative binomial regression I guess they performed a case-complete analysis? N=431 (for WEEKDAY) and N=396 (for WEEKEND) in Table 3 are case-complete sample sizes? Please, clarify. As limits of the study, authors should mention the fact they did not measure IMD, BMI, physical activity etc. in the two time points. As important limit of the study, authors have to add the fact they have only 685 children included in both phases.
---

VERSION 1 – AUTHOR RESPONSE

Reviewer: 1

The authors of "Socio-economic patterns of changes in children's screen-viewing between ages 6 and 9: a longitudinal study" address the question of how socioeconomic differences influence changes in screen-viewing time among children between 6 and 9 years.

I have the following remarks for the authors to consider:

Major:

The content of the paper is not easy to follow. The presentation of the results is only partial, e.g.

increases of screen viewing time is described only for three out of four socioeconomic subgroups. The figures (especially Figure 2) are not very comprehensible, too.

We have restructured all parts of the paper to focus more clearly on the main aim. The results section now focuses on socio-economic differences between all four groups (for example, lines 216-217) In addition, we have removed Figure 2, and replaced Figure 1 with a better figure to highlight the main socio-economic differences more clearly.

Title:

What do the authors mean with "socioeconomic patterns"? Why don't they use "status". They don't describe any patterns.

The title does not correspond well with the aim described in the introduction.

We have amended the title to 'Associations between socio-economic position and changes in children's screen-viewing between ages 6 and 9: a longitudinal study' to better reflect the aim of the paper.

Abstract:

Please don't use abbreviations in the abstract or at least explain them when used for the first time. The authors present results in the conclusion section that they did not mention in the results section. Please change that.

We have amended the results section and conclusion to simplify the presentation and interpretation while also staying within the word count (lines 29-36). This was achieved by focussing on the overall pattern of findings. Please see the revised abstract.

Introduction:

The introduction section is very short with weak information about the existing body of knowledge. (Percentage of screen time of overall sedentary time in comparable population groups, recommendations regarding sitting behaviours...) The authors mix sedentary behaviour with screen viewing time and cite results of studies that don't fit very well into the context. (p6, 110-14).

We have expanded the introduction to include more details of the background and relevant research (lines 66-77), and to focus specifically on screen-viewing and socio-economic differences (lines 91-103). While screen viewing is of course a type of sedentary behaviour, we have removed some of the sedentary time references, and added in more references to screen-viewing specifically to shift the focus.

Methods:

Please explain the educational levels. If abbreviations are used for the first time, they have to be explained. Please explain "GCSE", "NVQ" and "HND".

We have given these abbreviations in full (lines 139-40) but retained the explanations (eg 'qualifications typical obtained at age 16') to suit an international audience.

Were height and weight measured at school or by the parents?

They were measured at school by trained fieldworkers – we have added this to the methods section (line 149-50)

Results:

The first main result is that screen viewing is higher among children of parents with lower education. This is described in Table 1 and 2 where it is comprehensible. It is also described in Figure 1, if I understand correctly. Please explain though, what you presented with Figure 2. The title is again "total screen viewing", however it is not explained what the authors mean with "modelled screen viewing". Are these changes of screen viewing? Where are the numbers presented in this figure in the text? For example: Figure 1 (right) shows about 240minutes of screen time (weekend) at age 9 in the GCSE group. Figure 2 (left) shows 190 minutes of screen time (weekend) at age 9 in the GCSE group. Please explain that. Titles of figure 1 and 2 are misleading, please clarify.

We have replaced Figure 1 with a better figure that illustrates the main result that screen viewing is higher among children of parents with lower education, and more easily allows comparison of education groups and the two time points. The new Figure 1 now clearly shows screen-viewing time for ages 6 and age 9, and the difference between the two.

Figure 2 was intended to illustrate the change in screen-viewing between ages 6 and 9, as described by the model in the old Table 3. However, we appreciate that the presented figure was confusing and not very useful, and so we have removed it.

The authors present only the changes of the A Level group and the university degree group compared to the GCSE group without mentioning the higher degree group. Why is that?

Initially we highlighted those differences that were statistically significant, although we appreciate that we should have stated there were no differences between GCSE and Higher Degree groups.

We have now rewritten this section to focus more clearly on whether socioeconomic differences are present and if they can be explained by other factors. This includes a discussion of all four education groups (lines 216-17, 227-28)

I think it is worth being mentioned that between the group with the lowest and the highest education level there were no differences in screen time changes. Please go more into detail when presenting these results.

We have rewritten the Results section to present more details and focus on socioeconomic differences throughout. In the new models, there were differences in screen time changes for the highest group, which we present at line 226-28.

Discussion:

When describing the main results, please mention all education groups!

We have described all the differences for each education group at lines 252-53.

Page 17, line 50: The presented results belong to the results section.

We have removed this from the discussion section.

Page 18, line 8: the number of devices changed in two directions. Less computers but more tablets. Please differentiate.

We have added this at lines 271-72.

Page 18, line 15: The authors present a rate reduction of screen time for having any tablets at home. Thus, the conclusion that the presented results are consistent with previous studies (number of devices is linked with screen time) is false. The result of the present study is contradictory to that.

We have rewritten this paragraph (lines 266-81) to give a more nuanced discussion of the relationship between screen-viewing and device ownership (lines 274-79), highlighting the extent to which our results are and are not consistent with other studies.

The authors discuss different activities that tablets can be used for, however, it was not the aim of the study to differentiate between different screen activities but between different devices.

We believe that different screen-viewing activities (with or without tablets) are still of interest, but as this is not the main focus of the paper, we have reduced the discussion of this point to give it less emphasis, and restrict it to suggesting possibilities for future research (lines 298-304).

Overall, the structure of the manuscript is not straight forward and should be revised.

Thank you for your helpful comments; we hope that the revised structure makes the paper much clearer to follow.

Reviewer: 2

First of all, I would like to thank you for the opportunity to review the manuscript entitled "Socioeconomic patterns of changes in children's screen-viewing between ages 6 and 9: a longitudinal study". The manuscript assesses different types of screen viewing in the context of age and socio-economic differences based on data from B-PROACT1V longitudinal study. Generally, I like the idea of the paper, however, I have several comments dealing with the study that should be addressed. I hope it will be helpful for improving the overall quality of the paper.

General comments

I have 3 main issues that I think should be addressed:

- I would recommend simplifying general storyline dealing with the main aim, results and discussion section. Sometimes it was difficult to follow and recognize main messages

Thank you for these comments. As described above, we have re-structured and simplified the manuscript throughout to focus on the key messages; we hope it is now clearer.

- I would emphasize and discuss more in detail the issue of not including tablets and smartphones

We have expanded on this limitation in the discussion (lines 295-98).

- It was not clear to me how multiscreen viewing was assessed or calculated

Multiscreen viewing was parent-reported, on the same scale as other types of screen viewing. We have now mentioned multiscreen viewing explicitly in the Methods section (lines 132-34)

so the method of assessment is clear, and added some comments in the limitations (lines 317-19).

Introduction

General comment

I would recommend adding to the text why the period between 6 and 9 is important for developing/fixing screen-based behaviour

Thanks for the suggestion. New text has been added to lines 67-77.

Page6/line9-18

I do not clearly understand the storyline of the introduction here with the ref. no 16. I think this part should be better structured.

As noted above, we have completely restructured the introduction, which hopefully addresses this comment. This specific reference (still [16]) now appears in context describing the inverse relationship between SES and sedentary time generally (line 93) before also discussing the relationship with sedentary behaviour and screen-viewing specifically.

Methods

Page6/line47

Could you please add the No. and the date of ethical approval?

This has now been added to line 114.

Page6-7

I would recommend adding more detailed information about the response rate. For example, what was the response rate on school level in Phase 1? What was the response rate on an individual level, how many children refused to participate? What were the main reasons for not participating in the study?

As this is part of a larger study, details of the recruitment process, flow diagram and response rates etc have been described elsewhere. In the interests of space, rather than repeat them here, we have provided a reference to the earlier papers (line 111-12).

Page7/8-11

Could you please add the information about sex, I think this should be mentioned in the text not only in tables?

We have added the sentence 'Child gender was reported by the first parent' at line 149.

page7/line26

Could you please explain how multiscreen viewing was assessed or calculated? The study of Jago 2011 that you refer to (ref. 26) stated that "there are currently no means of assessing multi-screen viewing and therefore techniques that assess multi-screen viewing are urgently needed". It seems to clarify this in the study.

See earlier comment on multiscreen viewing, especially a mention of the limitations of this at line 317-19.

page7/line47-52

In terms of the international character of the journal I would recommend explaining the abbreviation dealing with the household education qualification

We have given these abbreviations in full (lines 139-40), but retained the explanations (eg 'qualifications typical obtained at age 16') to suit an international audience.

Page9/line 26-34 Imputation – I would consider to state only for example: “no imputed data were included in the study” and delete the rest

We have removed this unnecessary detail as suggested.

Results

Page9/line 50-55

I would not recommend starting the first sentence of the results section like this – please consider to change it. Moreover, considering the main aim, I don't think that the storyline of the results should start with the differences between weekdays and weekend.

We have restructured the results section to better reflect the focus of the paper. Consequently, we now begin discussion of the results with differences in screen viewing between education groups (line 187-96).

Page 9/first paragraph

Please consider adding the information about the age like... “At the age of 6... children...

We have rewritten the results section, and included cross-sectional associations at age 6. We have made clear at each point whether results refer to age 6, age 9 or the change between the ages.

Page10/line8

I would recommend skipping the sentence: The percentage of sedentary...

We have removed this sentence, and the corresponding lines from the table, to simplify the presentation and focus more clearly on screen-viewing rather than sedentary time/behaviour more generally.

Discussion

I would recommend to follow the same storyline from the results section and change the discussion accordingly.

The discussion has been completely rewritten; we hope this addresses your comment.

Limitations

Authors stated that the main limitation of the study is the absence of assessing “tablet time”. It seems to me that also measuring time spent on smartphones might be important for the results and should be mentioned in the limitation of the study. I would also recommend that the potential influence of the results dealing with this should be better discussed.

We agree that not accounting for smartphones might also affect results. At this age, smartphone use is more likely to be app-based similar to tablet use, and so we have combined discussion of these two, and have added this to the discussion (lines 294-97).

Abstract

Please change according to previous comments.

The Abstract has been rewritten to reflect the new structure of the paper, following reviewer comments.

Reviewer: 3

The authors stated they “...not presented any imputed data in the paper”, because they encountered problems with convergence in applying multiple imputation methods. However, it is not clear how they consequently dealt with missing data. Since they used multivariable negative binomial regression I guess they performed a case-complete analysis? N=431 (for WEEKDAY) and N=396 (for WEEKEND) in Table 3 are case-complete sample sizes? Please, clarify.

We did indeed use complete case analysis – we have now made this clear in the methods (line 181).

As limits of the study, authors should mention the fact they did not measure IMD, BMI, physical activity etc. in the two time points.

We measured all these variables at both time points as reported in old Tables 1 & 2 (now S1 and S2).

The model we presented in the original paper was for screen viewing at age 9, adjusting for baseline screen viewing at age 6 – in this model we included BMI etc measured at age 9 rather than baseline measurements.

As well as adding a cross-sectional model for age 6 (Table 1; lines 209-223), we have now described all the models in more detail in the methods section so it is clear what variables are included at what time points (specifically, at lines 219 and 230).

As important limit of the study, authors have to add the fact they have only 685 children included in both phases.

We have clearly stated the numbers included in the analysis in the methods (lines 122-23), and response rates etc described in more detail in the referenced original study (line 111-12 references 30-32).

While it is important to consider follow-up rates in any longitudinal study, we feel that our sample size is acceptable for a cluster design and does not constitute a major limitation to the study. For example, there are still 100-250 children in each of the education groups of interest. Moreover, as we have included SES directly, our models will not be biased due to differential follow-up rates by SES. We have added this to the limitations at lines 319-22.

VERSION 2 – REVIEW

REVIEWER	Lilian Krist Charité-Universitätsmedizin Berlin, Germany
REVIEW RETURNED	23-Jul-2019

GENERAL COMMENTS	Thank you very much for the revised version of your manuscript "Associations between socio-economic position and changes in children's screen-viewing between ages 6 and 9: a longitudinal study". The authors added some new information to the manuscript and tried to make it clearer. Unfortunately, there are still remaining a lot of parts in the manuscript which is not easy to understand. I have two major concerns:  1. The conclusion that socio-economic status (represented by education) is associated with screen time is not correct. In the adjusted model all associations don't reach statistical significance. However, the authors report mostly the unadjusted results which is misleading. Please describe the adjusted results which show, that it is unlikely that the education but simply the ownership of devices is associated with education. 2. The discussion is missing the most important part, namely the comparison with other studies. The authors repeat the results section and add explanations to their results, however, a manuscript cannot add to the scientific knowledge if the results are not considered into the context of those. Please refer to at least 10 other studies in your research field to compare your results with the existing knowledge. Some other points: Page 5, Line 21: the limit is 2 hours. Page 7 Line 27: are the analyses presented in Table 2 performed including all students or only the 685 students who participated at both time points? Please explain. If you included all students, you cannot call the study a cohort study, since you mixed it up with cross-sectional study design. Figure 1: Confidence Intervals are overlapping. The authors should interpret the differences between education groups with caution! Figure 2: The figure shows the contrary to the text ("..., while Higher Degree households had more computers"). The figure shows the Higher Degree group having the lowest number of computers. Results, page 11, line 20ff: the authors mention model 2, but don't describe the adjusted results. Same for model 3 and 4. Table 1 and 2:
--

	Why do the authors show p-values if they report also confidence intervals? If they do so, why are several p-values missing? Table 2: Why is an increase of 111 to 133 (22 minutes, 19.8%) in the A level group resulting in a higher OR (1.13) than an increase of 93-113 (20 minutes, 21.5%) in the Degree group (OR 1.06)? Please consider a revision of your analyses or please explain the way you came to your results. How have you calculated your p-values? If the confidence interval is (0.77;1.05) a p-value of 0.008 is impossible. Other examples: (0.83;1.10, p=0.004) or (0.87;1.11, p=0.006)
--	---

REVIEWER	Zdenek Hamrik Department of Recreation and Leisure Studies, Faculty of Physical Culture, Palacky University Olomouc, Czech Republic
REVIEW RETURNED	02-Aug-2019

GENERAL COMMENTS	I would like to thank the authors for addressing my previous comments and making all the necessary amendments. I have only 2 minor additional comments:  - Results section - line 210 - I would recommend reformulating the paragraph skipping the sentence "...which should be interpreted..." I think tables should be in some ways self-explanatory so I do not like this sentence here and would prefer to focus only on the description of the results. - I would recommend being more concrete and state more practical implications of the research in conclusions
--

REVIEWER	Augusto Filippo Di Castelnuovo Mediterranea Cardiocentro, Napoli, Italy
REVIEW RETURNED	09-Jul-2019

GENERAL COMMENTS	The authors provided a satisfactory revision
--

VERSION 2 – AUTHOR RESPONSE

Reviewer: 1

Reviewer Name: Lilian Krist

Thank you very much for the revised version of your manuscript "Associations between socio-economic position and changes in children's screen-viewing between ages 6 and 9: a longitudinal study".

The authors added some new information to the manuscript and tried to make it clearer. Unfortunately, there are still remaining a lot of parts in the manuscript which is not easy to understand.

I have two major concerns:

1. The conclusion that socio-economic status (represented by education) is associated with screen time is not correct. In the adjusted model all associations don't reach statistical significance. However, the authors report mostly the unadjusted results which is misleading.

Please describe the adjusted results which show, that it is unlikely that the education but simply the ownership of devices is associated with education.

Education is a categorical variable and so we have reported both individual estimates for each education category with confidence intervals, plus an overall p-value for the hypothesis test with the null hypothesis that all education categories are equal. Leaving aside a discussion of whether it is appropriate to focus overly on statistical significance in an observational study, the overall p-value should be used to determine whether education is statistically significant in these models, and not the individual confidence intervals. Thus, there are no significant differences between education categories in the adjusted models at age 6 (Table 1), but there are differences in the adjusted model for the change between ages 6 and 9 (Table 2 – p-values are 0.008 and 0.006 for weekday and weekend screen-viewing respectively). Thus, we believe that our results section is an accurate representation of the statistical model.

In our results we have discussed both the unadjusted and adjusted models. We have used the unadjusted models to report general patterns in screen-viewing between education groups (at lines 214-7 for age 6, and lines 224-9 for change between ages 6 and 9) as describing these differences is of interest in its own right. We have also reported the adjusted models to see whether these differences can be attributed to other factors such as numbers of devices (lines 218-223 for age 6 and lines 230-234 for change between ages 6 and 9). In particular, we have noted when the adjusted model no longer shows evidence of an association with education: 'These factors accounted for household education differences on both weekdays and weekends' (line 219-20). Where there is evidence of remaining education differences, we have used the estimate and confidence intervals to pick out general patterns (lines 230-1).

We disagree that the conclusion of our analysis is that 'the ownership of devices is associated with education'. We believe the correct interpretation of our results is that differences in screen-viewing that are present at age 6 are likely to be due to factors such as ownership of devices, but that change in screen-viewing between ages 6 and 9 differs by education, and cannot be explained by changes in device ownership alone. This is reflected in our discussion.

We have made some clarification changes relating to this misunderstanding based on your specific comments below, and address those points in more detail. We hope these changes make the results clearer.

2. The discussion is missing the most important part, namely the comparison with other studies. The authors repeat the results section and add explanations to their results, however, a manuscript cannot add to the scientific knowledge if the results are not considered into the context of those.

Please refer to at least 10 other studies in your research field to compare your results with the existing knowledge.

We have added a new paragraph lines 305-321 that provides greater discussion in relation to the literature. We have focussed this text on the key studies that impact on findings.

Some other points:

Page 5, Line 21: the limit is 2 hours.

As noted in the introduction (lines 66-67) the AAP's most recent advice (see reference) recommends that families develop a 'personalized media use plan', which replaces the older two-hour limit.

Page 7 Line 27: are the analyses presented in Table 2 performed including all students or only the 685 students who participated at both time points? Please explain. If you included all students, you cannot call the study a cohort study, since you mixed it up with cross-sectional study design.

The analysis in Table 2 (change between age 6 and age 9) is indeed based on the 685 children with data at both time points, as it would not be possible to fit this model unless this were the case. Are you referring to Table 1 which is a cross-sectional analysis of the 1299 children with data at age 6?

The B-Proact1v study itself is a cohort study as we have collected data on the same children at multiple time points. As part of this study, the current paper reports a cross-sectional analysis and a cohort (longitudinal) analysis, both of which are clearly labelled. Different types of analysis do not change the nature of the underlying study, so it is possible to report a cross-sectional analysis from a cohort study.

We have added N to each table to make the sample size at each point clearer, and we have added the words 'cross-sectional' and 'longitudinal' to the titles of Tables 1 and 2 respectively.

Figure 1: Confidence Intervals are overlapping. The authors should interpret the differences between education groups with caution!

The Reviewer is mistaken, Figure 1 does not show confidence intervals. As described in the title 'Boxplots of children's total minutes screen-viewing by household education at ages 6 and 9' these are boxplots of the raw data, which illustrate general patterns and the extent of the variability.

Figure 2: The figure shows the contrary to the text ("..., while Higher Degree households had more computers"). The figure shows the Higher Degree group having the lowest number of computers.

You are correct – the figure has the labels switched around by mistake. We have corrected the figure so that it now matches the text.

Results, page 11, line 20ff: the authors mention model 2, but don't describe the adjusted results. Same for model 3 and 4.

Model 2 is the adjusted model. The paragraph referred to here discusses Model 1, the unadjusted model at lines 213-217, and Model 2, the adjusted model, at lines 217-222. Similarly, the following paragraph describes Model 3, the unadjusted model, at lines 223-28 and Model 4, the adjusted model, at lines 229-33.

We have added additional signposting to these sections (lines 213, 217, 223, 224, 229) to make it clear which results correspond to the different models.

Table 1 and 2:

Why do the authors show p-values if they report also confidence intervals? If they do so, why are several p-values missing?

Confidence intervals and p-values are not interchangeable, and capture different things. Note that it is not really appropriate to use CIs to determine statistical significance. We have thus used the p-value to determine whether education categories are equal, which relates directly to the main aim of the paper. We also report the confidence intervals as it of interest to see the size and variability in the estimates.

There are no p-values missing from the tables; the categorical variable education has a single p-value. We have added a note to the table to make it clear that this p-value refers to the hypothesis tests of whether the education categories are equal.

Table 2: Why is an increase of 111 to 133 (22 minutes, 19.8%) in the A level group resulting in a higher OR (1.13) than an increase of 93-113 (20 minutes, 21.5%) in the Degree group (OR 1.06)?

The numbers you quote (111 mins and 133 mins for A level at ages 6 and 9 respectively, and 93 and 113 mins for the Degree group) come from Tables S1 & S2, which are the raw cross-sectional means. The ORs reported in Table 2 are from a longitudinal negative binomial model that adjusts for IMD, and clustering within schools. They are different because you are not comparing like with like. Specifically, the model links the same child at ages 6 and 9, and thus some of the differences are likely to be due to within-child versus between-child variability.

Please consider a revision of your analyses or please explain the way you came to your results.

We have made changes in response to your comments to add signposting and additional notes to clarify the results.

How have you calculated your p-values? If the confidence interval is (0.77;1.05) a p-value of 0.008 is impossible. Other examples: (0.83;1.10, p=0.004) or (0.87;1.11, p=0.006)

As explained above, the p-value does not relate to a single education category, but to an overall test of equality.

Reviewer: 2

I would like to thank the authors for addressing my previous comments and making all the necessary amendments. I have only 2 minor additional comments:

- Results section - line 210 - I would recommend reformulating the paragraph skipping the sentence "...which should be interpreted..." I think tables should be in some ways self-explanatory so I do not like this sentence here and would prefer to focus only on the description of the results.

We have removed this sentence as suggested.

- I would recommend being more concrete and state more practical implications of the research in conclusions

We have now added a specific suggestion to the conclusion on lines 351-353.

Reviewer: 3

The authors provided a satisfactory revision